# Monitoring of the Pre-Equilibrium Step in the Alkyne Hydration Reaction Catalyzed by Au(III) Complexes: A Computational Study Based on Experimental Evidences

**DOI:** 10.3390/molecules26092445

**Published:** 2021-04-22

**Authors:** Flavio Sabatelli, Jacopo Segato, Leonardo Belpassi, Alessandro Del Zotto, Daniele Zuccaccia, Paola Belanzoni

**Affiliations:** 1Dipartimento di Chimica, Biologia e Biotecnologie, Università degli Studi di Perugia, Via Elce di Sotto 8, I-06123 Perugia, Italy; flavio.sabatelli@gmail.com; 2Dipartimento di Scienze Agroalimentari, Ambientali e Animali, Sezione di Chimica, Università di Udine, Via Cotonificio 108, I-33100 Udine, Italy; segato.jacopo@spes.uniud.it (J.S.); alessandro.delzotto@uniud.it (A.D.Z.); 3Istituto di Scienze e Tecnologie Chimiche del CNR “Giulio Natta” (CNR-SCITEC), 00185 Rome, Italy; leonardo.belpassi@cnr.it

**Keywords:** gold(III) complexes, alkyne hydration, pre-equilibrium, coordination ability, density functional theory, catalyst design

## Abstract

The coordination ability of the [(ppy)Au(IPr)]^2+^ fragment [ppy = 2-phenylpyridine, IPr = 1,3-*bis*(2,6-di-isopropylphenyl)-imidazol-2-ylidene] towards different anionic and neutral X ligands (X = Cl^−^, BF_4_^−^, OTf^−^, H_2_O, 2-butyne, 3-hexyne) commonly involved in the crucial pre-equilibrium step of the alkyne hydration reaction is computationally investigated to shed light on unexpected experimental observations on its catalytic activity. Experiment reveals that BF_4_^−^ and OTf^−^ have very similar coordination ability towards [(ppy)Au(IPr)]^2+^ and slightly less than water, whereas the alkyne complex could not be observed in solution at least at the NMR sensitivity. Due to the steric hindrance/dispersion interaction balance between X and IPr, the [(ppy)Au(IPr)]^2+^ fragment is computationally found to be much less selective than a model [(ppy)Au(NHC)]^2+^ (NHC = 1,3-dimethylimidazol-2-ylidene) fragment towards the different ligands, in particular OTf^−^ and BF_4_^−^, in agreement with experiment. Effect of the ancillary ligand substitution demonstrates that the coordination ability of Au(III) is quantitatively strongly affected by the nature of the ligands (even more than the net charge of the complex) and that all the investigated gold fragments coordinate to alkynes more strongly than H_2_O. Remarkably, a stabilization of the water-coordinating species with respect to the alkyne-coordinating one can only be achieved within a microsolvation model, which reconciles theory with experiment. All the results reported here suggest that both the Au(III) fragment coordination ability and its proper computational modelling in the experimental conditions are fundamental issues for the design of efficient catalysts.

## 1. Introduction

Compared to Au(I) [1,2,3,4,5,6,7,8,9,10,11], catalysis by Au(III) is far less developed. The study of Au(III) chemistry is much more challenging both experimentally and theoretically, because these complexes are usually very reactive and difficult to synthesize. However, in recent years, many studies on the catalytic efficiency of Au(III) have been carried out, motivated by the expected more efficient activation of double and triple C-C bonds due to the larger Lewis acidity of the metal in its +3 oxidation state [12,13,14,15,16,17,18,19,20,21,22,23,24,25]. First studies were devoted to synthesize stable Au(III) complexes. In fact, the high reduction potential of Au(III) makes it very unstable in solution. Au(III) easily tends to reduce to Au(I) or Au(0) if electron-rich species are present in the reaction environment [26,27,28,29,30,31,32,33]. On the other hand, the ligands that completely stabilize the charge of the metal generate a catalytically non-active complex [34]. For example, the oxidative addition product, L-AuX_3_, formed from L-AuX (X = halogen) acts as a poorly efficient catalyst and is easily reduced. Moreover, while N-heterocyclic carbene gold(III) complexes can be prepared through multi-step synthetic sequences [35,36], their instability in the cationic form has severely limited their applications in catalysis. The problem has been solved with the use of pincer ligands to stabilize Au(III), while maintaining its catalytic activity. These ligands are able to stabilize *bis*- and mono-cyclometalated gold(III) complexes, avoiding reductive elimination to Au(I) or Au(0) [37]. The most used ligands and their synthetic strategy to generate mono- or *bis*-cyclometalated Au(III) complexes are summarized in the review by Nevado and Kumar [38]. In this field, a theoretical study by some of us on the alkyne activation with Au(III) complexes with different pincer ligands has shown that the Au(III)-alkyne complexes very efficiently activate the alkyne triple bond to the point that the nucleophilic attack step could cease to be the rate determining step (RDS) of the reaction, at variance with Au(I)-alkyne complexes [39]. Then, other steps of the catalytic cycle, particularly the pre-equilibrium was suggested to be a key step for the reaction. The mechanism of the hydration of 3-hexyne catalyzed by [(ppy)Au(IPr)Cl]Cl (**1**) [ppy = 2-phenylpyridine, IPr = 1,3-*bis*(2,6-di-isopropylphenyl)-imidazol-2-ylidene] in γ-valerolactone (GVL) as the solvent has been investigated by some of us both experimentally (NMR) and computationally (DFT), demonstrating that the pre-equilibrium step is effectively the RDS. As a matter of fact, water or counterion substitution by 3-hexyne in the first co-ordination sphere of Au(III) is crucial for the whole process [40]. The hydration of alkynes is an important reaction in organic chemistry, and it turns to be one of the most environmentally friendly methods to form the C=O bond [41,42]. All these findings suggest that the coordination ability of Au(III) complexes is a fundamental issue for the design of increasingly performing catalysts. Despite these achievements, much remains to be further explored. Gold(III) alkyne complexes have still to be crystallographically characterized and very little is known about the role of ancillary ligands in gold(III) chemistry until now. 

The aim of this work is exactly to computationally investigate the coordination ability of catalytic Au(III) complexes and its dependence on the ancillary ligands on the basis of some unexpected experimental evidences acquired by studying the alkyne hydration reaction. As mentioned, the coordination ability of the gold(III) monocationic/bicationic fragments towards the counterion, alkyne or nucleophile has an impact on the pre-equilibrium step of the catalytic cycle, favoring it or deactivating the catalyst, as demonstrated for a gold(I) monocationic fragment [43,44,45,46,47,48].

On the basis of the here reported experimental data, the coordinating ability of the [(ppy)Au(IPr)]^2+^ fragment towards Cl^−^, BF_4_^−^, OTf^−^, H_2_O, 3-hexyne, and 2-butyne has been computationally studied. Moreover, the effect of the ancillary ligand modelling, namely, replacing of IPr with NHC, and of the ancillary ligand substitution, namely, replacing of ppy and IPr with (C^N^C) (C^N^C = 2,6-*bis*(4-*^t^*BuC_6_H_3_)_2_ pyridine dianion) and of IPr with Cl^−^, on Au(III) coordination ability has been investigated. Although in computational studies the ancillary ligands are commonly simplified, when one explores the energetics of ion pair dissociation, which are strongly governed by steric factors, a simplified model of the ligand could be no longer acceptable. All the results are compared with those obtained for the [(NHC)Au]^+^ fragment, in order to highlight differences and similarities between the gold +1/+3 oxidation states. Finally, the pre-equilibrium step has been studied by evaluating the activation energy for the H_2_O substitution by 2-butyne in [(ppy)Au(NHC)(H_2_O)]^2+^ and [(NHC)Au(H_2_O)]^+^ complexes under modelled experimental conditions, namely, in the presence of explicit GVL and H_2_O molecules as the solvents.

## 2. Computational Details

DFT calculations have been performed using the Amsterdam Density Functional (ADF) (2016 version) [49,50,51] and the related Quantum-regions Interconnected by Local Descriptions (QUILD) [52] program packages. For geometry optimization, the GGA BP86 functional [53,54] with inclusion of the Grimme 3 BJDAMP dispersion correction [55] (BP86-D3) and the dichloromethane as the solvent by the Conductor like Screening Model COSMO [56,57,58], were employed in combination with a Slater-type TZ2P triple zeta basis set with two polarization functions for all atoms, in the small frozen core approximation. Relativistic effects were included by the scalar zero-order regular approximation ZORA Hamiltonian [59,60,61]. Frequency calculations have been performed at the same level of theory to compute bonding free energies and to identify all stationary points as minima (zero imaginary frequencies) or transition states (one imaginary frequency). All calculations were carried out for the closed-shell singlet state. A detailed methodological study, where dispersion and solvation effects on the bonding energies and geometries of the considered complexes are explicitly evaluated, can be found in the Appendix A. 

## 3. Results and Discussion

### 3.1. Experimental Evidence on the Coordination Ability of [(ppy)Au(IPr)]^2+^

The catalytic activity of [(ppy)Au(IPr)Cl]Cl (**1**) in the 3-hexyne hydration reaction has shown a not completely satisfactory efficiency of this complex [40].

However, in an attempt to clarify key aspects of the mechanism and to characterize and isolate the reaction intermediates of the pre-equilibrium step of the alkyne hydration reaction mechanism, namely the complexes [(ppy)Au(IPr)(3-hexyne)]^2+^ (**2**), [(ppy)Au(IPr)H_2_O]^2+^ (**3**), [(ppy)Au(IPr)OTf]^+^ (**4**), and [(ppy)Au(IPr)BF_4_]^+^ (**5**), the reactivity of **1** towards the addition of AgBF_4_ or AgOTf in different solvents, in the presence of 3-hexyne or water was studied. This exploration allowed to gain useful insight into the coordination ability of the [(ppy)Au(IPr)]^2+^ fragment towards the different species present in the catalytic conditions. Relevant findings are briefly described below (see Scheme 1). The starting complex (or pre-catalyst) **1** coordinates the chloride ions in the first and second coordination shells. Cl^−^ must be removed in order to coordinate and activate the 3-hexyne substrate. Experimentally, this step requires the addition of silver salts, and therefore AgOTf and AgBF_4_ were used to remove the Cl^−^ ions in different conditions (Scheme 1). [(ppy)Au(IPr)H_2_O](BF_4_)_2_ (**3BF_4_**) and [(ppy)Au(IPr)BF_4_]BF_4_ (**5BF_4_**) complexes were generated in a NMR tube by reacting **1** with 2 equiv of AgBF_4_ in non-anhydrous CD_2_Cl_2_ (for experimental study details see Appendix A). From ^1^H, ^13^C, ^1^H-COSY, ^1^H-NOESY, ^1^H,^13^C-HMQC NMR, and ^1^H,^13^C-HMBC NMR spectroscopies all proton and carbon resonances belonging to the different fragments of both complexes were assigned. The two complexes are present in about equal amount (52% of **5BF_4_**) and monitoring the chemical exchange between the two resonances of proton H1 by qualitative ^1^H-EXSY NMR experiments allowed to extract the rate constant (k_obs_) for their interconversion, k_obs_ was found to be 1.0 s^−1^.

In order to synthesize the complex [(ppy)Au(IPr)(3-hexyne)](BF_4_)_2_ (**2BF_4_**), 10 equiv. of 3-hexyne were added to the CD_2_Cl_2_ solution containing **3BF_4_** and **5BF_4_**, but no variations of the resonances belonging to both complexes were observed, thus indicating that no formation of alkyne complex **2BF_4_** was attained. On the contrary, the addition of 5 equiv. of water quantitatively shifted the **3BF_4_**/**5BF_4_** equilibrium towards the former.

Then, we decided to try to synthesize [(ppy)Au(IPr)H_2_O](OTf)_2_ (**3OTf**) and [(ppy)Au(IPr)OTf]OTf (**4OTf**) complexes by following the same procedure, but the formation of [(ppy)Au(IPr)Cl]OTf (**6OTf**) was obtained (Scheme 1). The addition of 4 equiv. of AgOTf to a solution of 1 in acetone at 50 °C overnight gave the formation of a mixture of **6OTf**, **4OTf**, and **3OTf** in 0.22, 0.70, 0.08 ratio (see Appendix A for details). Analogously in this case, the addition of 10 equiv. of 3-hexyne to a CD_2_Cl_2_ solution of the mixture did not change its composition, while the addition of water gave the formation of a mixture of **6OTf** and **3OTf**. 

For a direct comparison with the mixture of **3BF_4_**/**5BF_4_** in dichloromethane, the NMR spectrum of the mixture **4OTf**/**3OTf** was recorded in the same solvent. A similar composition of both mixtures was observed indicating a similar coordination behavior of BF_4_^−^ and OTf^−^ towards the gold fragment (see Appendix A).

In summary, the species **2** was not detected in our NMR experiments when BF_4_^−^, OTf^−^ or water are present into the mixture. In addition, BF_4_^−^ and OTf^−^ showed the same coordinating ability towards [(ppy)Au(IPr)]^2+^ and slightly less than water. To date, only the water-containing species **3BF_4_** can be isolated and characterized, consistently with the highly reactive character of the catalyst-substrate complex during the catalysis [39].

These results contrast with what found for instance for the [(NHC)Au]^+^ (NHC = 1,3-dimethylimidazol-2-ylidene) fragment, where 3-hexyne and OTf^−^ are more coordinating than water and BF_4_^−^ [43,44,45,46,47,48]. As a partial confirmation, Glorius and coworkers have isolated a similar compound, [(ppy)Au(PPh_3_)BF_4_]^+^, in which the BF_4_^−^ ion is coordinated in the inner sphere instead of water [62]. 

These experimental counterintuitive results and challenges have been addressed computationally in this work in an attempt to rationalize them and, eventually, to suggest how to improve the catalytic efficiency of this type of Au(III) complexes.

### 3.2. Coordination Ability of the [(ppy)Au(IPr)]^2+^ Fragment

To study the coordination ability of the [(ppy)Au(IPr)]^2+^ fragment, the bonding energies of X (X = Cl^−^, BF_4_^−^, OTf^−^, H_2_O, 2-butyne and 3-hexyne) have been evaluated. We have performed calculations of electronic energy (∆E), enthalpy (∆H) and Gibbs free energy (ΔG) change for the following reaction:[(ppy)Au(IPr)]^2+^ + X^−/0^ → [(ppy)Au(IPr)X]^+/2+^(1)

The optimized geometrical structures of all the systems are reported in the Appendix A (Appendix A).

As the X ligands are concerned, we selected all the experimentally employed species. In addition to 3-hexyne, also 2-butyne was examined, which is usually considered as a model alkyne in computational investigations. The results are summarized in Table 1.

The bonding free energies (∆G) span a range of −0.4/−44.0 kcal/mol and, as a general trend, we can immediately see that the coordination ability decreases in the order of Cl^−^ > OTf^−^ > 3-hexyne > 2-butyne ≈ BF_4_^−^ > H_2_O. Note that the same qualitative trend can be found considering the ∆E and ∆H values. Remarkably, ∆G values for OTf^−^, BF_4_^−^, 3-hexyne and 2-butyne are very close, in the range of −12.5/−17.5 kcal/mol, thus suggesting that they should have a similar coordination ability towards [(ppy)Au(IPr)]^2+^or, rather, that the [(ppy)Au(IPr)]^2+^ fragment should be poorly selective towards these four ligands, irrespective of their different charge. The Cl^−^ strongest coordination ability is in full agreement with the difficulty experimentally encountered when attempts are made to remove the chloride from the first coordination shell of the Au(III) complex. By contrast, removal of Cl^−^ from L-Au-Cl by AgX is experimentally easier [47]. Results reported in Appendix A in the Appendix A show however that free energy for chloride binding to Au(III) is very similar to that to Au(I) (−44.0 vs. −43.9 kcal/mol, respectively). Then, other effects should be taken into account to rationalize this finding. On the computational side, for instance, the continuum implicit solvent model we are using here could not be completely satisfactory for comparing Au(III)-Au(I) coordination ability to chloride at the actual experimental conditions (for a discussion on this issue, see Section 3.4).

On comparing the coordination ability of BF_4_^−^ and OTf^−^ to the metal center, a difference of only 5.0 kcal/mol is calculated, with OTf^−^ being more coordinating. This relatively low value is unexpected on the basis of the commonly recognized, both theoretically and experimentally, higher coordination ability of OTf^−^ compared to BF_4_^−^. However, this finding is fully consistent with the experimentally observed similar coordination ability of BF_4_^−^ and OTf^−^ towards the [(ppy)Au(IPr)]^2+^ fragment, which is only slightly shifted in favour of the triflate ion. We can surmise that the large steric hindrance due to the two isopropylphenyl groups of IPr is responsible for a larger destabilization of the Au(III)-OTf bond than the Au(III)-BF_4_ one caused by the larger size and less “spherical” symmetry of OTf^−^ compared to BF_4_^−^. As a consequence, the coordination ability of OTf^−^ decreases in the [(ppy)Au(IPr)X]^+^ complex, although remaining slightly more coordinating than BF_4_^−^. However, the decreasing contribution of the steric hindrance to the OTf^−^ bonding energy could be counteracted by an additional effect still due to the IPr and ppy ligands, namely the long-range noncovalent interactions, which are demonstrated to play an important role in these large dimension systems where high polarization of the ligands fragments is expected [63,64]. In the optimized structures of the [(ppy)Au(IPr)X]^+/2+^ (X = Cl^−^, OTf^−^, BF_4_^−^, 3-hexyne, 2-butyne, and H_2_O) complexes which are compared in Appendix A in the Appendix A, the nearest contacts between the ligand X and the hydrogens from the ppy ring and the isopropylphenyl groups of IPr are highlighted.

Precisely in order to investigate this issue, the coordination ability trend in a [(ppy)Au(NHC)]^2+^ (NHC = 1,3-dimethylimidazol-2-ylidene) fragment, with a simplified IPr ligand (the two isopropylphenyl moieties are replaced by methyl groups), commonly used in computational studies on IPr-based gold complexes, is analyzed. A comparative analysis of the experimental vs. model fragment coordination ability is presented in the Appendix A (Appendix A, Appendix A). The bonding energies of the [(ppy)Au(NHC)]^2+^ fragment generally increase for all the X ligands with respect to those of [(ppy)Au(IPr)]^2+^ (by 6-15 kcal/mol) and the [(ppy)Au(NHC)]^2+^ fragment is found to be more selective towards OTf^−^ than BF_4_^−^ (the difference in ∆G increases from 5 kcal/mol in [(ppy)Au(IPr)]^2+^ to 10 kcal/mol in [(ppy)Au(NHC)]^2+^). Overall, these results suggest that the most important contribution to the OTf^−^ bonding energy in the [(ppy)Au(IPr)]^2+^ fragment is much more likely the destabilizing steric effect of IPr.

Focusing on neutral ligand series, as a balance between the steric effects due to the IPr ligand and the stabilizing effects of long-range noncovalent interactions between alkyne and IPr and ppy, both the alkynes are found to be more strongly coordinating to Au than water, with larger size 3-hexyne coordinating better than 2-butyne (see Table 1), thus indicating that the steric hindrance should not be the only effect that contributes to the selectivity of the metal towards the considered neutral ligands. Remarkably, H_2_O exhibits the smallest calculated bonding free energy in both the [(ppy)Au(IPr)H_2_O]^2+^ and [(ppy)Au(NHC)H_2_O]^2+^ complexes (Table 1, Appendix A and Appendix A). This finding, which is in strong disagreement with the experiment, suggests that the [(ppy)Au(IPr)]^2+^ complex under study would be more π-philic than oxophilic, at variance with common assumption in the literature that gold(III) catalysts do have a strong oxophilic nature, whereas gold(I) catalysts do not [17,65,66,67]. A comparison with the coordination ability of a [(NHC)Au]^+^ fragment towards X = Cl^−^, OTf^−^, BF_4_^−^, 3-hexyne, 2-butyne, H_2_O has been also reported in the Appendix A (Appendix A, Appendix A), showing that, analogously, H_2_O is the weakest ligand in the whole series. In conclusion, due to the steric hindrance/dispersion interaction balance between X and IPr, the [(ppy)Au(IPr)]^2+^ fragment is found to be less selective towards the different ligands, with OTf^−^ and BF_4_^−^ showing very similar coordination strength, thus rationalizing the experimentally observed equilibrium between **4OTf** and **3OTf** and, analogously, between **5BF_4_** and **3BF_4_**. On the contrary, this factor cannot explain the stability of the water-containing species **3BF_4_** which could be experimentally isolated and characterized and vice versa the failure in isolating the alkyne complexes. The coordination ability has been evaluated using an implicit continuum solvation model which may be not fully satisfactory to establish an accurate scale of coordination ability at the actual catalytic conditions. Before addressing this issue, in the following section we investigate the effect of the ancillary ligand on the coordination ability of the metal fragment towards alkynes and water.

### 3.3. Ligand Effect on the Coordination Ability: Alkynes vs. Water

To shed light into the above intriguing results for neutral ligands, where H_2_O results a weaker coordinating ligand than alkynes, the ancillary ligand influence on the Au(III) coordination ability is now analyzed. 

Among suitable ancillary ligands, we selected the *bis*-cyclometalated [(C^N^C)Au]^+^ (C^N^C = 2,6-*bis*(4-*^t^*BuC_6_H_3_)_2_ pyridine dianion) and the monocyclometalated [(ppy)AuCl]^+^ Au(III) complexes. In particular, alkyne complexes with [(C^N^C)Au(III)]^+^ have been experimentally observed and characterized by Bochmann et al. with different internal alkynes [68,69]. Theoretical studies on the [(C^N^C)Au(III)]^+^ alkyne and carbonyl complexes have also been performed previously by some of us [39,70]. Instead, the [(ppy)AuCl]^+^ complex, where a Cl^−^ replaces the IPr ligand in our reference complex, allows for a direct comparison with [(ppy)Au(IPr)]^2+^, where the X interactions with IPr are switched off. In addition, the [(ppy)Au(NHC)]^2+^ complex has been considered to test the commonly used procedure of modelling the IPr with NHC in computational studies and the Au(I) complex, [(NHC)Au]^+^ is examined, for a comparison with Au(III). The optimized structures of [(C^N^C)AuX]^+^ and [(ppy)AuClX]^+^ (X = H_2_O, 2-butyne, 3-hexyne) are shown in the Appendix A (Appendix A). 

Analogously to [(ppy)Au(IPr)]^2+^, the bonding free energies of X (X = H_2_O, 2-butyne, 3-hexyne) have been calculated for the following reactions:[(C^N^C)Au]^+^ + X → [(C^N^C)AuX]^+^(2)
[(ppy)AuCl]^+^ + X → [(ppy)AuClX]^+^(3)
[(ppy)Au(NHC)]^2+^ + X → [(ppy)Au(NHC)X]^2+^(4)
[(NHC)Au]^+^ + X → [(NHC)AuX]^+^(5)

The overall energetic trend in all the systems can be readily visualized in Figure 1. The numerical data have been reported in the Appendix A (Appendix A).

From Appendix A and Figure 1 we can observe that, although the coordination ability of Au(III) is quantitatively strongly affected by the ligands, the coordination ability trend is the same for all the fragments, namely 3-hexyne > 2-butyne > H_2_O. In particular, for a given X (X = H_2_O, 3-hexyne, 2-butyne), the coordination ability follows the trend [(C^N^C)Au]^+^ > [(NHC)Au]^+^ > [(ppy)Au(NHC)]^2+^ > [(ppy)AuCl]^+^ > [(ppy)Au(IPr)]^2+^. One dominant factor influencing the strength of the Au-X interaction might be the trans influence of the ancillary ligand [68]. In [(C^N^C)AuX]^+^ the X ligand feels only the relatively weak effect of a trans pyridine, whereas in [(ppy)AuClX]^+^, [(ppy)Au(NHC)X]^2+^ and [(ppy)Au(IPr)X]^2+^ the X is trans to a phenyl group with a much stronger effect. Notably, only Au(III) in [(C^N^C)Au]^+^ can coordinate alkynes better than the [(NHC)Au]^+^ fragment. In addition, these results suggest that Au(III) coordination ability does not depend on the charge of the complex: for example, [(C^N^C)Au]^+^ coordinates to X more strongly than [(ppy)Au(NHC)]^2+^ with 2+ charge which, in turn, coordinates to X more strongly than [(ppy)AuCl]^+^. Again, these findings seem to do not support a commonly encountered postulate in the literature according to which Au(III) species are more oxophilic in nature whereas Au(I) species show a more π-philic property [17,65,66], since, within the considered auxiliary ligands set, both Au(III) and Au(I) show a stronger coordination ability towards alkynes than water, at least within an implicit solvent model. Finally, on comparing [(ppy)Au(NHC)]^2+^, [(ppy)AuCl]^+^ and [(ppy)Au(IPr)]^2+^, replacing of IPr with NHC or Cl^−^ has a beneficial effect on coordination ability, thus reasserting a sizable role of IPr steric interactions with X. 

### 3.4. Substitution of H_2_O With the Substrate: The Pre-Equilibrium Step

From the above study, water clearly emerges as the weakest ligand in the whole series and significantly weaker than alkynes. All the studied Au(III) complexes are important species in catalytic alkyne nucleophilic addition reaction. The first step of the reaction mechanism is represented by the pre-equilibrium, where the substrate (alkyne) replaces the anion in the initial complex, binds to the gold center, and is activated for the nucleophilic attack. However, in the alkyne hydration reaction, where H_2_O is the nucleophile, the anion substitution by the water molecule, i.e., formation of the water adduct from the corresponding initial complex, has been shown to be both thermodynamically and kinetically favored with respect to the formation of alkyne complex, in agreement with the experimental evidence (see also Introduction and Scheme 1 for details) [40]. Therefore, starting from the experimentally isolated and characterized **3**, we will explore the pre-equilibrium step consisting of H_2_O substitution by the alkyne and rationalize the reason why **2** is not experimentally observed. The experimental conditions of the alkyne hydration reaction need to be clearly modelled beyond the implicit solvation model we have employed above. In a reaction environment where anionic ligands are not close to the Au(III) catalytic complex (for instance, using highly polar solvents or very weakly coordinating anions) the present species can be additional water, alkyne and/or high polarity solvent molecules. To model such reaction conditions, calculations of the initial complex (IC) and reactant complex (RC) using the Au(III) model complex and 2-butyne have been performed in an attempt to study the pre-equilibrium step of the hydration reaction mechanism. The IC is represented by the Au(III)-H_2_O model complex with 2-butyne in the second coordination sphere, whereas the RC is the Au(III)-2-butyne model complex with H_2_O in the second coordination sphere. To account for the experimental conditions, the IC and RC calculations have been performed including: i) an additional water molecule to model traces of water or water solvent effect; ii) a GVL molecule to model the highly polar aprotic solvent effect actually used in the experimental study. We should remark that these calculations have been performed with exactly the same computational protocol as that used for the coordination ability study, namely by including the continuum solvent model (COSMO) with dichloromethane as the solvent, for a direct comparison with data in Appendix A in the Appendix A. The results are summarized in Figure 2, where the energy profiles for the pre-equilibrium step have been shown. 

Interestingly, in both cases IC is found to be more stable than RC, thus indicating that water is a stronger ligand than 2-butyne if we include in our model the microsolvation effect, which is more closely related with the actual experimental conditions. It is eye-catching that the IC stabilization is due to an explicit interaction of the coordinated H_2_O with a solvent molecule, namely that only microsolvation can account for “oxophilicity” of gold(III) which could not be found using an implicit solvent model (COSMO calculations). The same result has been also found in the gas phase (no solvation model), where the aquo complex IC is more stable than the 2-butyne complex RC by 6.4–4.5 kcal/mol (see corresponding Appendix A in the Appendix A). Transition state calculations for the IC → RC process show that the free energy activation barrier is 6.0 kcal/mol for H_2_O and 4.2 kcal/mol for GVL. However, the reverse process RC → IC requires a lower barrier (only 2.0 kcal/mol for GVL and 3.7 kcal/mol for H_2_O). Our calculations clearly point out that IC is thermodynamically more stable than RC, which rationalizes the difficulty to isolate and characterize the alkyne complex in the experimental conditions: anytime the complex binds to the alkyne, it is substituted by the H_2_O.

The presence of a polar molecule (H_2_O or GVL) which is able to establish a hydrogen interaction with water in IC_H2O_ or IC_GVL_ polarizes the water O-H bond, increasing the partial negative charge on the oxygen atom which becomes more coordinating toward the metal center. The lower activation energy barrier obtained when the solvent is GVL, compared to water, suggests that the “activation” of the O-H bond is prevalently caused by the hydrogen bond strength. Indeed, in IC_GVL_ structure, the water O-H distance involved in GVL hydrogen bond is 1.029 Å, whereas in IC_H2O_ structure the corresponding O-H distance is 1.026 Å, thus indicating that GVL is more efficient in “activating” H_2_O to an incipient “OH^−^”.

A direct comparison with the Au(I) catalyst, [(NHC)Au]^+^ is reported in Figure 2. The most noticeable difference relies on the more stable RC compared to IC. Therefore, in the Au(I) case, water is less strong than 2-butyne as ligand, and an equilibrium shift towards the RC species is suggested. The partial negative charge on the oxygen atom of the water O-H bond induced by microsolvation appears to be less stabilized by gold +1 charge than +3, as one could expect. A more realistic description of the experimental conditions would however require a sample of the conformational space with inclusion of a large number of explicit water or GVL molecules. These calculations are outside the scope of the present paper. On the other hand, our simple model allows for a detailed microscopic rationalization for experimental findings which interestingly suggests that microsolvation could be very important for the pre-equilibrium step of the [(ppy)Au(IPr)]^2+^–catalysed hydration of alkynes and which calls for further investigations on this issue in other Au(III)-catalyzed reaction types. 

## 4. Conclusions

The pre-equilibrium is a key step in the Au(III)-catalyzed alkyne hydration reaction [40]. Despite its importance, the Au(III) complexes affinity towards different species involved in the catalytic cycle (water, counterions and alkynes) is far to be understood. In this work the binding energy of the [(ppy)Au(IPr)]^2+^ fragment [ppy = 2-phenylpyridine, IPr = 1,3-*bis*(2,6-di-isopropylphenyl)-imidazol-2-ylidene] towards different anionic and neutral X ligands (X = Cl^−^, BF_4_^−^, OTf^−^, H_2_O, alkynes) is computationally investigated to shed light on unexpected experimental observations on its catalytic activity in the alkyne hydration reaction. Experimental evidences for a BF_4_^−^ and OTf^−^ very similar coordination ability towards [(ppy)Au(IPr)]^2+^ and slightly less than water are given. An equilibrium between **4OTf** and **3OTf** and, analogously, between **5BF_4_** and **3BF_4_** was observed, while the alkyne complex could not be observed in solution at least at the NMR sensitivity. 

The computational study shows that, due to the steric hindrance/dispersion interaction balance between X and IPr, the [(ppy)Au(IPr)]^2+^ fragment is much less selective than a simplified model [(ppy)Au(NHC)]^2+^ (NHC = 1,3-dimethylimidazol-2-ylidene) fragment towards OTf^−^ and BF_4_^−^, which exhibit very similar coordination strength, in agreement with the experiment. The higher steric hindrance in [(ppy)Au(IPr)]^2+^due to the two isopropylphenyl groups of IPr is responsible for a larger destabilization of the Au(III)-OTf bond than the Au(III)-BF_4_ one, caused by the larger size and less “spherical” symmetry of OTf^−^ with respect to BF_4_^−^, thus suggesting that both qualitatively and quantitatively the ligand modelling does affect the coordination ability of this IPr-Au type fragment. 

Effect of the ancillary ligand substitution, by considering the [(C^N^C)Au]^+^ (C^N^C = 2,6-*bis*(4-*^t^*BuC_6_H_3_)_2_ pyridine dianion) and [(ppy)AuCl]^+^ fragments, and comparison with a typical Au(I) complex, [(NHC)Au]^+^, demonstrates that i) the coordination ability of Au(III) is quantitatively strongly affected by the ligands (even more than the net charge of the complex) according to the [(C^N^C)Au]^+^ > [(ppy)Au(NHC)]^2+^ > [(ppy)AuCl]^+^ > [(ppy)Au(IPr)]^2+^ trend, with the Au(I) complex being in between [(C^N^C)Au]^+^ and [(ppy)Au(NHC)]^2+^ ii) the coordination ability trend is the same for all the complexes, namely 3-hexyne > 2-butyne > H_2_O within the neutral ligand subset. These results seem not to support a common postulate present in the literature, according to which the Au(III) species are more “oxophilic” in nature whereas the Au(I) species are more “carbophilic”, at least within the implicit solvent model. Among the X ligands, H_2_O results in the weakest ligand in the series in all the investigated Au(III), Au(III) model and Au(I) complexes, in strong disagreement with the experimental evidence. However, experiment and theory have been reconciled through proper modelling of the experimental catalytic environment by including explicit GVL and H_2_O molecules, respectively, as the solvent. Only within this microsolvation model, the water-containing complex is more stable than the alkyne-containing one. This study advises that the experimental conditions of the pre-equilibrium step are crucial in Au(III) catalysis as well as their proper computational modelling. Thus, to experimentally isolate and characterize the [(ppy)Au(IPr)(alkyne)]^2+^ complex, an anhydrous environment is required combined with the use of very non-coordinating anions [71], which means that alkyne hydration reaction is not a suitable process to reach this goal. Other Au(III)-catalyzed reactions, such as hydroarylation of alkynes, Meyer-Schuster rearrangement of propalgyl alcohols or even alkoxylation of alkynes, could be more appropriate for this target.

## Data Availability

Data supporting reported results are available online.

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
