# Peer review of "Monitoring of the Pre-Equilibrium Step in the Alkyne Hydration Reaction Catalyzed by Au(III) Complexes: A Computational Study Based on Experimental Evidences"

_molecules, 2021, doi:10.3390/molecules26092445_

Round 1
Reviewer 1 Report
In this paper the authors report the outcomes of combined experimental and computational investigation to shed light on the coordination ability of the [(ppy)Au(IPr)]2+ fragment [ppy = 2-phenylpyridine, IPr =1,3-bis(2,6-di-isopropylphenyl)-imidazol-2-ylidene] towards different anionic and neutral X ligands (X = Cl-, BF4-, OTf-, H2O, 2-butyne, 3-hexyne) commonly involved in the crucial preequilibrium step of the alkyne hydration reaction. Effects of the substituents on the key N-heterocyclic carbene ligand and of the ancillary ligands on the Au(III) center, as well as of changing Au(III) to Au(I), have been investigated and the results discussed in view of the catalytic activity of this class of complexes.
The investigated system is of interest in the promising field of gold(III) catalysis, and the theoretical approach is carried out at an adequate level of theory, given the size of the considered systems.
However, the paper suffers of few problems which should be overcome before the paper can be accepted for publication:
- Page 4, line 8 from the bottom “In addition, BF4- and OTf- showed the same coordinating ability towards [(ppy)Au(IPr)]2+ and...”. Actually, in the same less polar CH2Cl2 solvent OTf- seems to be less coordinating as it does not displace chloride from Au(III) while BF4- does. OTf- can displace chloride only in the more polar acetone solvent.
- Page 5, paragraph 3.2. The solvent considered in the calculations on the coordinating ability of the six considered ligands is not explicitly stated, although it is probably dichloromethane. However, binding to Au(III) of the OTf- ligand has been experimentally observed only in acetone; this point deserves a comment if not further calculations in acetone solvent.
- Page 5, paragraph 3.2, results in Table 1. For all the considered ligands, except chloride, the reaction free energy values are ca. 10-15 kcal/mol higher than reaction enthalpies, as expected for the coordination reaction (1) with a loss of a translational degree of freedom. On the other hand, for chloride Table 1 shows a decrease of 3.5 kcal/mol passing from reaction enthalpy to reaction free energy: this is quite anomalous and should be carefully checked and in case it is confirmed it should be discussed. I would expect a value around 30-35 kcal/mol, not 54.4 kcal/mol. How binding of a ligand to a metal center can occur with and entropy gain, and why only for chloride?
- Page 5, paragraph 3.2, 10 lines below Table 1. The sentence “By contrast, removal of Cl- from L-Au-Cl by AgX is more easier.” is probably experimentally true (although the reference has a wrong bookmark and cannot be checked) but is not proved by the theoretical calculations, and indeed the results in Table S1 (which should be discussed here) contradict it. Indeed, reaction free energy for chloride binding to Au(III) is similar to that to Au(I). By the way, “more” before “easier” could be deleted.
- Page 5, paragraph 3.2, 8 lines above the bottom. Commenting the theoretical finding of a better coordination ability of OTf- compared to BF4-, with a higher binding free energy of 5 kcal/mol) the authors write: “…this finding is fully consistent with the experimentally observed similar coordination ability of BF4- and OTf- towards the [(ppy)Au(IPr)]2+ fragment, which is only slightly shifted in favour of the triflate ion.”. However, from the experimental results reported in paragraph 3.1 and resumed in Scheme 1, it is clear that OTf- does not replace the coordinated chloride in dichloromethane while BF4- does, suggesting rather a worse coordination ability of OTf- compared to BF4-. Probably this theoretical finding should be better discussed in terms of the experimental results in Scheme 1. (see also point 1)
- Page 8-10, paragraph 3.4. Here it is shown that the inclusion of one explicit polar molecule simulating microsolvation (water or GVL) stabilizes H2O binding to [(ppy)Au(NHC)]2+ with respect to the binding of 2-butyne making the resulting aquo complex ca. 2 kcal/mol more stable in free energy, at variance with the results obtained from continuum solvent model leading instead to the 2-butyne complex more stable in free energy by ca 10 kcal/mol (Table S2). This is clearly stated at page 9, 3 lines below figure 2: “It is eye-catching that the IC stabilization is due to an explicit interaction of the coordinated H2O with a solvent molecule, namely that only microsolvation can account for “oxophi-licity” of gold(III) which could not be found using an implicit solvent model (COSMO calculations).” However, the authors are comparing explicit solvation using one H2O or GVL molecule with continuum solvent (COSMO) calculations using dichloromethane as solvent. So they are simultaneously changing not only the solvation model (explicit – implicit) but also the considered solvent (low polar dichloromethane with a dielectric constant of 8.9 – high polar solvent such water with dielectric constant of ca. 80) mixing the two effects. I think this point should be better discussed, including a more direct comparison of the results in Figure 2 with those in Table S1 and trying to disentangle the effects from the solvation model (explicit – implicit) and from the different polarity of the solvents.
- A few examples of recent DFT calculations on Au(I) NHC complexes could be cited, such as Tobatov et al. Insight into the Substitution Mechanism of Antitumor Au(I) N‑Heterocyclic Carbene Complexes by Cysteine and Selenocysteine Inorg. Chem. 2020, 59, 5, 3312–3320.
Author Response
The Reviewer writes:
In this paper the authors report the outcomes of combined experimental and computational investigation to shed light on the coordination ability of the [(ppy)Au(IPr)]2+ fragment [ppy = 2-phenylpyridine, IPr =1,3-bis(2,6-di-isopropylphenyl)-imidazol-2-ylidene] towards different anionic and neutral X ligands (X = Cl-, BF4-, OTf-, H2O, 2-butyne, 3-hexyne) commonly involved in the crucial preequilibrium step of the alkyne hydration reaction. Effects of the substituents on the key N-heterocyclic carbene ligand and of the ancillary ligands on the Au(III) center, as well as of changing Au(III) to Au(I), have been investigated and the results discussed in view of the catalytic activity of this class of complexes.
The investigated system is of interest in the promising field of gold(III) catalysis, and the theoretical approach is carried out at an adequate level of theory, given the size of the considered systems.
Author reply: We thank the Reviewer for appreciation of our work.
The Reviewer writes:
However, the paper suffers of few problems which should be overcome before the paper can be accepted for publication:
- Page 4, line 8 from the bottom “In addition, BF4- and OTf- showed the same coordinating ability towards [(ppy)Au(IPr)]2+ and...”. Actually, in the same less polar CH2Cl2 solvent OTf- seems to be less coordinating as it does not displace chloride from Au(III) while BF4- does. OTf- can displace chloride only in the more polar
acetone solvent.
Author reply: We thank the Reviewer for this comment, which allows us to clarify this point. We performed abstraction of chloride from 1 in acetone taking advantage of AgOTf, but the NMR spectrum of the mixture (4OTf/3OTf) was recorded in dichloromethane for a direct comparison with the mixture of 3BF4/5BF4. As stated in the paper, we observe a similar composition of both mixtures indicating a similar coordination behavior of BF4 and OTf in the same CH2Cl2 solvent towards gold in agreement with DFT calculations.
We have now specified it in the revised version of the paper, page 4, line 1 below Scheme 1: “For a direct comparison with the mixture of 3BF4/5BF4 in dichloromethane, the NMR spectrum of the mixture 4OTf/3OTf was recorded in the same solvent. A similar composition of both mixtures was observed indicating a similar coordination behavior of BF4- and OTf- towards the gold fragment (see Supporting Information).”
The Reviewer writes:
2. Page 5, paragraph 3.2. The solvent considered in the calculations on the coordinating ability of the six considered ligands is not explicitly stated, although it is probably dichloromethane. However, binding to Au(III) of the OTf- ligand has been experimentally observed only in acetone; this point deserves a comment if not further calculations in acetone solvent.
Author reply: Actually the solvent considered in the calculations is explicitly stated in the Computational Details section (page 3, section 2. line 5 “...dichloromethane as the solvent...”). We have now commented on the binding to Au(III) of the OTf- ligand in dichloromethane solvent (see response to issue 1 above).
The Reviewer writes:
3. Page 5, paragraph 3.2, results in Table 1. For all the considered ligands, except chloride, the reaction free energy values are ca. 10-15 kcal/mol higher than reaction enthalpies, as expected for the coordination reaction (1) with a loss of a translational degree of freedom. On the other hand, for chloride Table 1 shows a decrease of 3.5 kcal/mol passing from reaction enthalpy to reaction free energy: this is quite anomalous and should be carefully checked and in case it is confirmed it should be discussed. I would expect a value around 30-35 kcal/mol, not 54.4 kcal/mol. How binding of a ligand to a metal center can occur with and entropy gain, and why only for chloride?
Author reply: We thank the Reviewer for having pointed out this mistake. We have now included in the ΔG calculations the chloride entropic contribution evaluated using the Sackur–Tetrode equation and the new values for ΔH and ΔG in Table 1 are -51.5 kcal/mol and -44.0 kcal/mol, respectively. We have also corrected the corresponding data in Table S1 in the SI.
The Reviewer writes:
4. Page 5, paragraph 3.2, 10 lines below Table 1. The sentence “By contrast, removal of Cl- from L-Au-Cl by AgX is more easier.” is probably experimentally true (although the reference has a wrong bookmark and cannot be checked) but is not proved by the theoretical calculations, and indeed the results in Table S1 (which should be discussed here) contradict it. Indeed, reaction free energy for chloride binding to Au(III) is similar to that to Au(I). By the way, “more” before “easier” could be deleted.
Author reply: We have corrected the reference bookmark (now 11 e)) which refer to experimental results for removal of Cl- from L-Au-Cl by AgX. We have discussed the results in Table S1, page 5, line 6 from below: “Results reported in Table S1 in the SI show however that free energy for chloride binding to Au(III) is very similar to that to Au(I) (-44.0 vs. -43.9 kcal/mol, respectively). Then, other effects should be taken into account to rationalize this finding. On the computational side, for instance, the continuum implicit solvent model we are using here could not be completely satisfactory for comparing Au(III)-A(I) coordination ability to chloride at the actual experimental conditions (for a discussion on this issue, see paragraph 3.4).” “More” before “easier” has been deleted.
The Reviewer writes:
5. Page 5, paragraph 3.2, 8 lines above the bottom. Commenting the theoretical finding of a better coordination ability of OTf- compared to BF4-, with a higher binding free energy of 5 kcal/mol) the authors write: “…this finding is fully consistent with the experimentally observed similar coordination ability of BF4- and OTf- towards the [(ppy)Au(IPr)]2+ fragment, which is only slightly shifted in favour of the triflate ion.”. However, from the experimental results reported in paragraph 3.1 and resumed in Scheme 1, it is clear that OTf- does not replace the coordinated chloride in dichloromethane while BF4- does, suggesting rather a worse coordination ability of OTf- compared to BF4-. Probably this theoretical finding should be better discussed in terms of the experimental results in Scheme 1. (see also point 1)
Author reply: see response to point 1 above
The Reviewer writes:
6. Page 8-10, paragraph 3.4. Here it is shown that the inclusion of one explicit polar molecule simulating microsolvation (water or GVL) stabilizes H2O binding to [(ppy)Au(NHC)]2+ with respect to the binding of 2-butyne making the resulting aquo complex ca. 2 kcal/mol more stable in free energy, at variance with the results obtained from continuum solvent model leading instead to the 2-butyne complex more stable in free energy by ca 10 kcal/mol (Table S2). This is clearly stated at page 9, 3 lines below figure 2: “It is eye-catching that the IC stabilization is due to an explicit interaction of the coordinated H2O with a solvent molecule, namely that only microsolvation can account for “oxophi-licity” of gold(III) which could not be found using an implicit solvent model (COSMO calculations).” However, the authors are comparing explicit solvation using one H2O or GVL molecule with continuum solvent (COSMO) calculations using dichloromethane as solvent. So they are simultaneously changing not only the solvation model (explicit – implicit) but also the considered solvent (low polar dichloromethane with a dielectric constant of 8.9 – high polar solvent such water with dielectric constant of ca. 80) mixing the two effects. I think this point should be better discussed, including a more direct comparison of the results in Figure 2 with those in Table S1 and trying to disentangle the effects from the solvation model (explicit – implicit) and from the different polarity of the solvents.
Author reply: Actually the calculations in paragraph 3.4 with one explicit polar molecule simulating microsolvation (water or GVL) have been performed with exactly the same computational protocol as that used for the coordination ability study, namely by including the continuum solvent model (COSMO) with dichloromethane as the solvent. For this reason the results in Figure 2 and in Table S1 are directly comparable (same solvation model - implicit). In addition, in Figure S21 in the SI corresponding results in the gas phase are presented (no solvation model), where the aquo complex is also more stable with respect to the binding of 2-butyne by 6-5 kcal/mol. We have clarified it in the text: page 9, line 5 from below: “We should remark that these calculations have been performed with exactly the same computational protocol as that used for the coordination ability study, namely by including the continuum solvent model (COSMO) with dichloromethane as the solvent, for a direct comparison with data in Table S1 in the Supporting Information.”; page 10, line 7 below Figure 2: “The same result has been also found in the gas phase (no solvation model), where the aquo complex IC is more stable than the 2-butyne complex RC by 6-5 kcal/mol (see corresponding Figure S21 in the SI).”
The Reviewer writes:
7. A few examples of recent DFT calculations on Au(I) NHC complexes could be cited, such as Tobatov et al. Insight into the Substitution Mechanism of Antitumor Au(I) N‑Heterocyclic Carbene Complexes by Cysteine and Selenocysteine Inorg. Chem. 2020, 59, 5, 3312–3320.
Author reply: We have cited this reference in the Introduction (11 f)).
Reviewer 2 Report
The authors present a combined experimental and computational investigation of the coordination chemistry of gold(III) complexes containing NHC and ppy ligands. They model the pre-equilibrium step of a more complex reaction network, namely the hydration of alkynes.
According to my expertise, I am more interested into the DFT part. The impression is that the authors did a lot of computational work and considered all of the features that they need to include in the level of theory (dispersion, solvent, advanced level of theory). Interestingly, the authors point out that the size of their model system is directly influencing the outcome of the prediction regarding the coordination ability of water and hexyne / butyne. The facts are suitable and well organized for publication in the journal “Molecules”.
Just a small note: B2PLYP with dispersion corrections can be regarded as B2PLYP-D, used also by Grimme et al.
A few references are printed as “Error! Bookmark not defined.“, please check what is missing there.
Author Response
The Reviewer writes: The authors present a combined experimental and computational investigation of the coordination chemistry of gold(III) complexes containing NHC and ppy ligands. They model the pre-equilibrium step of a more complex reaction network, namely the hydration of alkynes.
According to my expertise, I am more interested into the DFT part. The impression is that the authors did a lot of computational work and considered all of the features that they need to include in the level of theory (dispersion, solvent, advanced level of theory). Interestingly, the authors point out that the size of their model system is directly influencing the outcome of the prediction regarding the coordination ability of water and hexyne / butyne. The facts are suitable and well organized for publication in the journal “Molecules”.
Author reply: We thank very much the Reviewer for the positive judgment of our work.
The Reviewer writes: Just a small note: B2PLYP with dispersion corrections can be regarded as B2PLYP-D, used also by Grimme et al.
Author reply: We have corrected it.
The Reviewer writes: A few references are printed as “Error! Bookmark not defined.“, please check what is missing there.
Author reply: We are sorry for this inconvenience. We have now corrected them.
Reviewer 3 Report
This manuscript describes coordination ability of a series of ligands and, in particular, alkynes to the gold(III) complexes. The work is well-written and the studied problem is very actual. In my opinion, this work should be accepted after minor revision accordingly to the following comments.
1) The authors provided an NMR experiment in the presence of 10-fold excess of the alkyne to the gold(III) complexes and found no peak shifts in the spectrum, but their theoretical calculations further indicated that difference between two forms (ligated H2O or alkyne) is ca. 2 kcal/mol. Thus, the equilibrium constant is potentially can be find from the NMR experiments. I recommend to provide NMR titration with higher ratios of the reagents or utilize UV spectroscopy to explore the K. Experimental calculation of relative energies of these forms is actual for comparing it with the theoretical results.
2) Table 1: for the chloride, free Gibbs energy is more negative than the enthalpy, which indicates increase of entropy of the system during coordination of the chloride. It is counterintuitive, because two species transforms to one. Probably, some error in computation occured. If no, some explanation of this phenomenon should be given in the main text.
P.7, line three from the bottom: delete a space in "Au (III)".
Author Response
The Reviewer writes: This manuscript describes coordination ability of a series of ligands and, in particular, alkynes to the gold(III) complexes. The work is well-written and the studied problem is very actual. In my opinion, this work should be accepted after minor revision accordingly to the following comments.
Author reply: We thank very much the Reviewer for the positive appreciation of our work.
The Reviewer writes: 1) The authors provided an NMR experiment in the presence of 10-fold excess of the alkyne to the gold(III) complexes and found no peak shifts in the spectrum, but their theoretical calculations further indicated that difference between two forms (ligated H2O or alkyne) is ca. 2 kcal/mol. Thus, the equilibrium constant is potentially can be find from the NMR experiments. I recommend to provide NMR titration with higher ratios of the reagents or utilize UV spectroscopy to explore the K. Experimental calculation of relative energies of these forms is actual for comparing it with the theoretical results.
Author reply: We thank the Reviewer for this comment, but unfortunately we don’t have an adequate UV spectrometer in our department facilities. By the way we performed an NMR experiment in the presence of 50-fold excess of the alkyne to the gold(III) complexes and we found only a broaden of the peaks of proton H1 but not appreciable shift.
The Reviewer writes: 2) Table 1: for the chloride, free Gibbs energy is more negative than the enthalpy, which indicates increase of entropy of the system during coordination of the chloride. It is counterintuitive, because two species transforms to one. Probably, some error in computation occured. If no, some explanation of this phenomenon should be given in the main text.
Author reply: We thank the Reviewer for having pointed out this error. We have now included in the ΔG calculations the chloride entropic contribution evaluated using the Sackur–Tetrode equation and the new values of ΔH and ΔG in Table 1 are -51.5 kcal/mol and -44.0 kcal/mol, respectively. We have also corrected the corresponding data in Table S1 in the SI.
The Reviewer writes: P.7, line three from the bottom: delete a space in "Au (III)".
Author reply: We have deleted it.
Round 2
Reviewer 1 Report
The authors have responded adequately to all the issues raised in my previous review, and I therefore consider the paper suitable to be published in Molecules as it is